



# Analysis of atmospheric particle growth based on vapor concentrations measured at the high-altitude GAW station Chacaltaya in the Bolivian Andes

Arto Heitto[1], Cheng Wu[2,a], Diego Aliaga[3], Luis Blacutt[4], Xuemeng Chen[3], Yvette Gramlich[2], Liine Heikkinen[2], Wei Huang[3], Radovan Krejci[2], Paolo Laj[3,5], Isabel Moreno[4], Karine Sellegri[6], Fernando Velarde[4], Kay Weinhold[7], Alfred Wiedensohler[7], Qiaozhi Zha[3], Federico Bianchi[3], Marcos Andrade[4], Kari E. J. Lehtinen[1], Claudia Mohr[2,b] and Taina Yli-Juuti[1]

[1]Department of Technical Physics, University of Eastern Finland, 70211 Kuopio, Finland
[2]Department of Environmental Science and Bolin Centre for Climate Research, Stockholm University, Stockholm, Sweden
[3]Institute for Atmospheric and Earth System Research/Physics, University of Helsinki, 00014 Helsinki, Finland
[4]Laboratorio de Física de la Atmósfera, Instituto de Investigaciones Físicas, Universidad Mayor de San Andrés, Bolivia
[5]Univ. Grenoble Alpes, CNRS, INRAE, IRD, Grenoble INP, IGE, 38000 Grenoble, France
[6]Université Clermont Auvergne, CNRS, LaMP, Clermont-Ferrand, France
[7]Leibniz Institute for Tropospheric Research

[a] now at: Department of Chemistry and Molecular Biology, University of Gothenburg, Gothenburg, Sweden
[b] now at: Laboratory of Atmospheric Chemistry, Paul Scherrer Institute, Villigen, Switzerland

*Correspondence to*: Arto Heitto (arto.heitto@uef.fi), Cheng Wu (cheng.wu@gu.se)

**Abstract.** Early growth of atmospheric particles is essential for their survival and ability to participate in cloud formation. Many different atmospheric vapors contribute to the growth, but even the main contributors still remain poorly identified in many environments, such as high-altitude sites. Based on measured organic vapor and sulfuric acid concentrations under ambient conditions, particle growth during new particle formation events was simulated and compared with the measured particle size distribution at Chacaltaya Global Atmosphere Watch station in Bolivia (5240 m a.s.l.) during April and May 2018,
as a part of the SALTENA (Southern Hemisphere high-ALTitude Experiment on particle Nucleation and growth) campaign . The simulations showed that the detected vapors were sufficient to explain the observed particle growth, although some discrepancies were found between modelled and measured particle growth rates. This study gives an insight on the key factors affecting the particle growth on the site. Low volatile organic compounds were found to be the main contributor to the particle growth, covering on average 65% of simulated particle mass in particle with diameter of 40 nm In addition, sulfuric acid had
a major contribution to the particle growth, covering at maximum 39% of simulated particle mass in 40 nm particle during periods when volcanic activity was detected on the area, suggesting that volcanic emissions can greatly enhance the particle growth.




## 1 Introduction

In favorable conditions, atmospheric particles can act as cloud condensation nuclei (CCN) and activate to form cloud droplets. It is estimated that up to half of the particles acting as CCN are secondary aerosol particles formed in the atmosphere by nucleation of oxidized gases (Merikanto et al., 2009). These secondary aerosol particles form when low-volatility vapors cluster and grow into particles of 1-2 nm in diameter (Kulmala et al., 2013). The formed particles are required to experience further growth to reach CCN sizes, which are typically at least some tens of nanometers in dry diameter (Pierce and Adams, 2007; Reddington et al., 2017). The fraction of newly formed particles that ends up growing to CCN sizes depends on the relative rates of particle growth and scavenging. The growth of particles serves as a source of CCN-sized particles while their loss due to collisions with pre-existing larger particles can be considered as a sink of them.

For the initial step of atmospheric new particle formation (NPF), i.e. nucleation, the concentration of sulfuric acid is of essence ( e.g. Kulmala and Kerminen, 2008). However, also ammonia, amines, and organic vapors are likely to partake in nucleation (Kulmala and Kerminen, 2008; Zhang et al., 2012, Kirkby et al., 2016). The nucleation mode particles grow when surrounding vapors condense on the particles. At least in areas dominated by biogenic emissions, the early growth is expected to be mostly governed by low and extremely low-volatile organic compounds (LVOC and ELVOC), although also semi-volatile organic compounds (SVOC), sulfuric acid, and ammonia may contribute to the growth if their vapor concentrations are high (Yli-Juuti et al., 2013; Ehn et al., 2014; Tröstl et al., 2016; Mohr et al., 2017; Kerminen et al., 2018; Mohr et al., 2019).

Three key features define regional atmospheric NPF: 1) Increase of particle concentration in nucleation mode (particles up to 25 nm in diameter), 2) formation of a new, distinct nucleation mode persisting for several hours, and 3) growth of the nucleation mode over several hours (Kulmala et al., 2012). If all these criteria are fulfilled continuously, it is characterized as an NPF event. NPF is a phenomenon frequently occurring in the atmosphere (Kulmala et al., 2004; Zhang et al., 2012; Kerminen et al., 2018). In previous studies, the growth rates (GR) in the nucleation mode have been shown to be in general around $1 - 20$ nm h$^{-1}$ (Yli-Juuti et al., 2011; Kerminen et al., 2018; Nieminen et al., 2018). The GR at high-altitude sites fall usually in the similar order of magnitude, although there is a large variability between different sites (Sellegri et al., 2019): greater GR have been found at the Southern Hemispheric high-altitude sites as opposed to high-altitude sites in the Northern Hemisphere. However, the observation locations discussed in Sellegri et al. (2019) are scarce: two data sets (about 1 year in span) are investigated from the Southern Hemisphere (Chacaltaya (5240 m a.s.l.) and Maido (2160 m a.s.l.) stations) and four stations from the Northern Hemisphere (Stations of Monte Cimone (2165 m a.s.l.), Jungfraujoch, (3580 m a.s.l.), Puy de Dôme (1465 m a.s.l.), and Nepal Climate Observatory Pyramid (5079 m a.s.l.) – length of time series was between one and four years). The pattern of higher GR at the Southern Hemisphere, as observed by Sellegri et al. (2019) for high-altitude sites, does not hold overall for all sites in the Northern and the Southern Hemisphere (Yli-Juuti et al., 2011; Nieminen et al., 2018). However, robust conclusions are hard to be drawn due to the limited data especially from the Southern Hemisphere (Kerminen et al., 2018; Laj et al., 2020). In addition to the scarcity of observational data, characterizing NPF events at high-altitude sites can be



very challenging for various reasons. One is the complicated topography of the sites, where different air masses and updrafts and mountain winds from the lower altitudes mix up the air, which can make the continuous observation of one NPF event very difficult (De Wekker and Kossman, 2015; Collaud Coen et al., 2018; Sellegri et al, 2019; Aliaga et al., 2021).

Here we analyze data from measurements that were conducted as part of the Southern Hemisphere high-ALTitude Experiment

on particle Nucleation and growth (SALTENA) on the Chacaltaya (CHC) Global Atmosphere Watch (GAW) station (5240 m a.s.l.; 16.35° S, 68.13° W) in Bolivia (Bianchi et al., 2022). CHC isinfluenced by free tropospheric and boundary layer air masses, often simultaneously (Aliaga et al., 2021). On average at any given time, about 24 % of the air arriving at Chacaltaya originates from the planetary boundary layer, and the other 76 % comes from the free troposphere (Aliaga et al., 2021). At Chacaltaya, multiple NPF events have been observed to take place on the same day consequently (Rose et al., 2015). In this

case, defining from which NPF event the nucleation mode particles originate may be difficult. In previous studies, the mean growth rates observed at the site were $10.31 \pm 14.65$ nm h$^{-1}$ and $13.65 \pm 15.91$ nm h$^{-1}$ in the particle diameter range of 3 - 7 nm and 7 - 20 nm, respectively (Sellegri et al., 2019). The GR at the site were generally higher during the wet season (November to April) than the dry season (May to October) (Rose et al., 2015). The measurement period analyzed in this work was within the transition of the seasons, during April and May. In addition, one specific feature of the observations at the measurement

site is the influence of volcanic emissions impacting at the site during the analyzed period in May, likely advected especially from Sabancaya volcano (Bianchi et al., 2021; Aliaga et al., 2021).

This work focuses on the growth of newly formed particles at CHC during the SALTENA campaign. The growth of secondary aerosol particles is simulated based on measured ambient gas-phase concentrations of sulfuric acid and organic compounds

and these simulations are compared with the observed particle growth during NPF events in April and May 2018. Our aim is to investigate whether condensation of the detected vapors can explain the observed growth of nucleation mode particles in this complex environment, the role different compounds play in the growth, and how other factors (i.e., ambient conditions) may contribute to the observed particle growth. As measuring the composition of the freshly formed nanoparticles currently remains a challenge, investigations based on gas-phase composition combined with modeling may provide a much-needed

increase in understanding the particle growth process.

## 2 Methods

The data utilized in this study were collected at the Chacaltaya GAW station during April and May 2018. The Chacaltaya Station is a high-altitude measurement station (lat. -16.350500°, lon. -68.131389°, 5240 m a.s.l.) on a slope of the Chacaltaya

mountain, which is part of a mountain ridge of the Bolivian Andes. The air masses arriving at the station have varying origins including the metropolitan area of La Paz, The Altiplano plane, the Pacific Ocean, and Amazonia (Chauvigné et al., 2019; Wiedensohler et al., 2019; Scholz et al 2023). Altogether, 36 NPF events were detected during the entire measurement period over April and May 2018. We restrict the analysis to 14 events (April 11, 15 to 17, 21 and 22, and May 10, 13, 22 and 26 to



30, see Table S1). The rest of the detected NPF events were left out of the analysis due to the lack of measurement data of one
or several of the most important input parameters for the model or due to problems with instrument calibration. To analyze
particle growth, we performed simulations with particle condensation model constrained by measured vapor concentrations
and compared the simulated particle growth with the observed growth based on particle size distribution measurements. By
using measured vapor concentrations, we are simulating how these vapors can grow the freshly formed particles.


## 2.1 Instrumentation

Oxygenated organic vapors were measured with a Filter Inlet for Gases and AEROsols (FIGAERO) coupled with a Time-of-
Flight Chemical Ionization Mass Spectrometer (ToF-CIMS) with iodide as reagent ion (Lopez-Hilfiker et al., 2014). The
FIGAERO-CIMS has two operational modes and monitors gas and particle-phase organic composition semi-continuously.
The duration of the gas-phase mode was 120 min, and it samples ambient air directly into the ion-molecular reactor (IMR).
During the gas-phase mode, particles are simultaneously collected on a 25 mm polytetrafluoroethylene (PTPE) filter through
another sampling port with a flow of 3.8 standard liter per minute (slpm). When the gas-phase measurement (particle
collection) is done, the FIGAERO inlet is switched to the particle desorption mode and a heated nitrogen gas flow (2 slpm) is
blown through the filter to evaporate the particle-phase compounds via temperature-programmed desorption. The duration of
the particle deposition period was 50 min. More details about the instrument can be found in Lopez-Hilfiker et al. (2014) and
Thornton et al. (2020). For each gas-phase measurement (120 minutes), we calculated four 30-min averaged data points.

For the conversion of the measured ion signal to mass concentrations, we took a "maximum sensitivity" of 20 counts per
second (cps) pptv$^{-1}$ (per million cps of reagent ion) from Lopez-Hilfiker et al. (2016), and adjusted that for the different IMR
pressure and flows going to the IMR in our measurements compared to the conditions in Lopez-Hilfiker et al. (2016). In our
study, the flow containing iodide ions was 1.3 slpm. Due to using a corona source (in April) and an X-ray ionizer (in May) as
ionization sources, the IMR pressure was set to 100 mbar in April and 480 mbar in May. The gas-phase sample flow was
controlled by a critical orifice. With an ambient pressure of about 530-540 mbar, the sample flow was about 1.1 slpm and 0.7
slpm in April and May, respectively. The ionizer and sample flows mix and interact in the IMR. The sensitivity due to the
changes in IMR pressure and the sampling flows (thus the residence time of reagent ion and analytes, and the time for their
reaction) was 30 and 140 cps pptv$^{-1}$ in April and May, respectively (for details see SI).

Gas-phase sulfuric acid concentration as well as organic compounds were measured with a nitrate-CIMS (Tofwerk AG, Thun,
Switzerland). The especially designed inlet for chemical ionization at ambient pressure is described by Kürten et al. (2011)
and Jokinen et al. (2012). The nitrate-CIMS uses nitrate anions [(HNO$_3$)n (NO$_3$- ), n = 0-2] as reagent ions to ionize gas
molecules. The sampling flow was 10 slpm, mixed with 20 slpm sheath flow before detection. The signal of the detected



compound in cps is normalized to the sum of count rates of reagent ions and then multiplied with the calibration coefficient $C$ = $1.5 \cdot 10^{10}$ cm$^{-3}$, which was determined after the campaign using $H_2SO_4$ as calibrant, following the procedure by Kürten et al. (2012). As the nitrate-CIMS was mainly sensitive to highly oxygenated organic molecules (HOMs), we compared the

organic compounds measured by the nitrate-CIMS and the FIGAERO iodide CIMS, and the absolute amount of organics detected by the nitrate-CIMS was much lower than that detected by FIGAERO CIMS, thus we only used the organic vapor concentrations from FIGAERO CIMS further for modelling (Fig. S1).

The particle number size distribution was measured using a Neutral cluster and Air Ion Spectrometer (NAIS, Mirme and

Mirme, 2011) and a Mobility Particle Size Spectrometer (MPSS, design TROPOS; Wiedensohler et al, 2018). The NAIS measured neutral and charged particles in the size range of 1-70 nm in diameter divided in 29 size bins. The time resolution of the NAIS data was 3.5 minutes. The MPSS detected particles in the size range 10-500 nm divided in 71 size bins with a time resolution of 2.5 minutes. The particle diameter in MPSS refers to dry size. As the particle sample transfers from outside ambient air to NAIS instrument, the temperature increase and RH decrease lead to evaporation of water. Due to lack of

information on the extend of water evaporation inside the instrument, the measured size in NAIS was also assumed to correspond to dry size in this study. In the analysis for acquiring the growth rates from measurements (see sect. 2.3), only NAIS data was utilized. For the rest of the analysis involving particle number size distributions, NAIS data for particles from 3 nm to 30 nm and MPSS data for particles larger than 30 nm were combined to obtain a size distribution for the size range of 3-500 nm.


An Automatic Weather Station (AWS) deployed in the main observatory recorded air temperature, relative humidity (RH), radiation, wind direction, and wind speed at a 1-minute resolution.

**2.2 Model**

The particle growth model used in our analysis is the Model for Oligomerization and Decomposition in NAnoparticle Growth (MODNAG) introduced by Heitto et al. (2021). In this study, however, we only use the condensation part of MODNAG and exclude the oligomerization and decomposition, since the particle phase reactions are omitted from this study for simplicity and due to the lack of data on relevant properties to constrain them. The model is a single aerosol particle condensation model,

where condensation of organic vapors, sulfuric acid, ammonia, and water are considered. The mass flux of each organic vapor and sulfuric acid between the gas and particle phase is calculated based on the difference of their ambient gas-phase concentrations and equilibrium vapor mass concentrations according to (Lehtinen and Kulmala, 2003)

$$\frac{dm_j}{dt} = 2\pi(d_p + d_j)(D_p + D_j)\beta_{m,j}(C_j - C_{eq,j}) \qquad (1)$$





where $d_j$ is the molecular diameter (m) and $D_j$ the gas-phase diffusion coefficient of the condensing compound j ($m^2 s^{-1}$), $C_j$ the

gas-phase mass concentration and $C_{eq,j}$ the equilibrium vapor mass concentration of compound j respectively (note that both

are here converted to units of kg $m^{-3}$), $d_p$ is the diameter (m) and $D_p$ the diffusion coefficient of the particle ($m^2 s^{-1}$). $\beta_i$ is the

transition regime correction factor. $C_{eq,j}$ is calculated based on solution effect and Kelvin effect as (e.g. Seinfeld and

Pandis,2016):

$$C_{eq,j} = \gamma_j \chi_j C_{sat,j_j} \exp\left(\frac{4\sigma v_j}{RTd_p}\right) \qquad (2)$$

where $\gamma_j$ is the activity coefficient, $\chi_j$ the mole fraction, $C_{sat,j}$ the pure compound saturation vapor concentration, $v_j$ the molar

volume of compound j, $\sigma$ the surface tension of the particle, R the gas constant and T the temperature. In this study we assume

an ideal solution and therefore the activity coefficient $\gamma_j$ is equal to 1.

$\beta_i$ in Eq. (1) is defined as (Fuchs and Sutugin, 1970):

$$\beta_j = \frac{1+Kn_j}{1+\left(\frac{4}{3\alpha_{m,j}}+0.377\right)Kn_j+\frac{4}{3\alpha_{m,j}}Kn_j^2} \qquad (3)$$

where $\alpha_{m,j}$ is the mass accommodation coefficient and $Kn_j$ is the Knudsen number. In this study, $\alpha_{m,j}$ is assumed to be unity for

all components. $Kn_j$ is defined as (Kulmala and Lehtinen, 2003):

$$Kn_j = \frac{2\lambda_j}{d_p+d_j} \qquad (4)$$

where $\lambda_j$ is the mean free path of the condensing molecule, which is calculated as

$$\lambda_j = \frac{3(D_p+D_j)}{\sqrt{c_p+c_j}} \qquad (5)$$

where $c_p$ and $c_j$ are the mean thermal speed of the particle and the condensing compound j, respectively.

In this study, we simulated a system that contains nine model compounds: six organic compounds (volatility bins), sulfuric

acid, water, and ammonia. The mass fluxes of organics and sulfuric acid between the gas and particle phases are calculated in

the model using Eq. (1). The particle water content is updated constantly by assuming that water is in equilibrium between the

gas and particle phases. The amount of ammonia in the particle phase is calculated by requiring it to match a 1:1 molar ratio

with sulfuric acid. This simplified assumption was necessary as gas-phase ammonia concentration data were not available.

Due to the small mass of the ammonia molecule, this assumption does not cause significant uncertainty in the simulation

results.

The six organic compounds are proxies for the measured organic vapors. For each measured organic compound, we assigned

a saturation vapor mass concentration ($C_{sat}$) value at 298 K based on their molecular formula using the volatility

parametrization by Li et al. (2016) and the temperature dependence of $C_{sat}$ by Epstein et al. (2010). Based on their volatilities,

the observed organic compounds were then divided into six groups and represented with the volatility basis set (VBS, Donahue

et al., 2006). The VBS bins ranged from $10^{-4}$ µg $m^{-3}$ to $10^1$ µg $m^{-3}$ at 298 K. All compounds with a $C_{sat}$ lower than $10^{-4}$ µg $m^{-3}$





(all ELVOCs) were included in the lowest volatility bin and compounds belonging to the bins with $C_{sat}$ higher than $10^1$ µg m$^{-3}$ (most SVOCs and everything more volatile) were neglected. This was done as our sensitivity tests showed that all compounds with $C_{sat}$ of $10^{-4}$ µg m$^{-3}$ or lower are effectively non-volatile from the point of view of the particle growth simulation in this study, and that the contribution of compounds with $C_{sat}$ higher than $10^1$ µg m$^{-3}$ to the particle growth is negligible. The sum of the concentrations of the vapors in each VBS bin was assigned for the concentration of the corresponding organic

model compound (i.e. VBS bin). The molar mass and gas-phase diffusion coefficients of the model compound were calculated as gas-phase concentration-weighted averages of the compounds belonging to the respective VBS bin.

The time resolution of the model input data was 30 minutes, i.e., in the beginning all measurement data was averaged over 30 minutes. For each simulation timestep, the properties and concentrations of organics, as well as ambient temperature, pressure,

and RH were linearly interpolated from the input data. For most of the time, a similar approach was used also for sulfuric acid. However, there were some gaps in the sulfuric acid data measured in April lasting over several hours. For these gaps in the data, the interpolation between measurements was not sufficient because of the strong diurnal variation of sulfuric acid concentrations. Instead, for these cases with missing sulfuric acid data (16$^{th}$ and 17$^{th}$ of April) we approximated the concentration with a Gaussian distribution. For this, first, the mean and standard deviation of the distribution were determined

by fitting a Gaussian distribution to normalized sulfuric acid concentrations of all NPF days in May with constant air mass (see section 2.3). Then the amplitude of the distribution for the day with gaps in the data was defined by fitting it to the available sulfuric acid data on that specific day. Illustration is shown in Figure S2a-b.

The initial particle size in the model was approximately 2 nm, containing 40 molecules of sulfuric acid and ammonia, and

particle-phase water corresponding to gas-particle equilibrium. This small size was selected to minimize the influence of assumed initial particle composition on the simulated growth. The first geometric mean diameters of growing nucleation mode were detected typically starting at around 7-10 nm, depending on the day; and in these sizes, with the selected small initial size, the initial composition covers less than 5% of the simulated particle mass. The choice of starting time of the model simulation was not straightforward. To define a suitable starting time for the simulation, a number of simulations were

performed with a range of starting times (with a 30-minute resolution). We then calculated the root-mean-square difference between the simulated particle diameter and the geometric mean diameter of the observed nucleation mode to find the best fit simulation. For this, only the geometric mean diameters up to 10 nm were considered to find the best fit for the beginning of the particle growth and to allow for a comparison of simulated and observed growth. On six days (April 16 and May 13, 26, 27, 28 and 30 of 2018) however, the simulated growth reached the 7-10 nm size range later than the observed geometric mean

diameter of the nucleation mode regardless of the starting time of the simulation, i.e., it was not possible to find a match between simulation and observations in this size range by adjusting the simulation starting time. In these cases, the gas-phase concentrations of organics and sulfuric acid were so low during the morning that the particle in the simulation grew to 7 nm very slowly. For these days we selected a simulation where the starting time was set at the time of sunrise, based on the





assumption that the nucleation and the initial growth are mostly governed by sulfuric acid, the concentration of which is highly
dependent on solar radiation (Mikkonen et al., 2011).

**2.3. Calculating growth rates from observations and model**

For calculating the particle diameter growth rates from observations, we used the method introduced by Dal Maso et al. (2005).
The GR values were derived from the size distribution data measured with the NAIS. In this method, first, the geometric mean
diameter of the nucleation mode at each timepoint is found by fitting a multi-modal log-normal size distribution to the measured
particle number size distribution. Then, a straight line is fitted to the data points of time and geometric mean diameter over the
growth period of the freshly nucleated particles to obtain the GR value. However, not all these data points were always included
when fitting the line for the GR as the geometric mean diameters did not always reflect a smoothly growing mode. Therefore,
linear fitting was performed on different subsets of these data points, selected by inspecting the size distribution evolution.
The fit that was estimated to best reflect the growing mode was selected as the base value and different versions of the fit
provided the uncertainty range. The GR values correspond mostly to the size range of 6-50 nm, however, the size range differs
between days depending on the part of the growing mode from which the GR value was obtained most reliably. Such an
approach was chosen as the aim was to compare observed and simulated particle growth.

From the modeled data, the GR was calculated from the change in the dry size of the particle (i.e., water excluded) during the
period corresponding to the experimentally defined growth rate. This means that for each fitted GR from the measurements, a
matching GR was defined from the models. When presenting the GR values, the base value of GR from the model is the GR
calculated from the same period as the base value from the measurements, and, analogous to measurements, the GR from other
periods provide the uncertainty range. On six days (May 10, 13, 22 and 28 to 30) two consecutive events were detected on the
same day. For these days, only the first event is included in the analysis due to the challenges of separating particles nucleated
from separate events.

The analysis of the growth during NPF events based on observations, requires – or includes an assumption – that the air mass
around the measurement site stays relatively homogeneous (regional NPF event). under stable atmospheric conditions). A
similar assumption is made when comparing the simulations, constrained by observed vapor concentrations, to the measured
growth of the particles. This, however, may not be the case for all of the events at the Chacaltaya station. The complex
topography of the surrounding area and the varying source areas of aerosols make it possible for air masses to change rapidly
in the area, and the site is often affected simultaneously by multiple air masses (Aliaga et al., 2021).

On four days (April 21 and 22, and May 26 and 27), around noon there was a sharp increase in the particle and organic vapor
concentrations, especially in SVOC and LVOC concentrations, as well as in black carbon (BC) concentrations. This
phenomenon was also reported by Aliaga et al. (2021) who concluded that this is an indication that a larger fraction of air mass





had been in contact with the surface in the previous few days. The increased BC concentrations during these high concentration periods (see Figure S3) further indicate that the air mass has been influenced by the La Paz/El Alto metropolitan area. On

forward these days will be addressed as days with inconstant or inhomogeneous air mass and the rest of the days will be addressed as days with constant or homogeneous air mass, respectively. We acknowledge that the air mass is probably not perfectly homogeneous all around the site in any of the studied events. However, the only subtle changes in measured particle number distributions and vapor concentrations imply that no drastic changes in airmass were occurring on events referred in this study as days with homogeneous air mass. In April and May, the influence of air mass with high particle and vapor

concentrations might last until the afternoon. In these cases, the changes in the particle size distribution observed at the station cannot be interpreted as continuous condensational growth of one particle population, and our model is not, in theory, expected to simulate the changes in particle size correctly. We applied our standard model simulations also for these days, however we performed also additional simulations for these cases to address the issue of air mass changes. By assuming that the particles observed before and after the period with elevated particle and vapor concentrations are from the same air mass, we performed

simulations for the growth during the NPF event by neglecting the measurements during the high-concentration period (i.e., surface-influenced air mass from the La Paz) and instead predicting what the conditions would have been in the original air mass, before the stark increase in particle and vapor concentrations. See SI for the details.


## 3. Results and discussion

Figure 1 shows the measured and modeled growth rates for each of the 14 NPF days included in the analysis. The figure shows with different markers the NPF events for which air masses were estimated to be fairly homogeneous throughout the growth

(filled markers) and the events for which changes in air mass were assumed to affect the particle size distribution evolution and measured gas phase concentrations (empty markers). The median and mean of measured GR for the entire size range (~ 5 – 60 nm, depending on the event, see Table S1) were 2.0 nm h$^{-1}$ and 2.6 nm h$^{-1}$ for all 14 days and 1.5 nm h$^{-1}$ and 1.7 nm h$^{-1}$ for days with homogeneous air mass, respectively. This is notably slower than the median GR for April and May (9.92 and 5.82 nm h$^{-1}$ for the size range of 3 – 7 nm, and 5.12 and 9.03 nm h$^{-1}$ for the size range of 7 - 20 nm, respectively) and the annual

mean GR (10.31 and 13.65 nm/h for size ranges of 3-7 nm and 7-20 nm, respectively) reported from the same site in previous studies (Rose et al., 2015; Sellegri et al., 2019). However, due to the limited dataset in our study, this is not necessarily an indication of discrepancy.



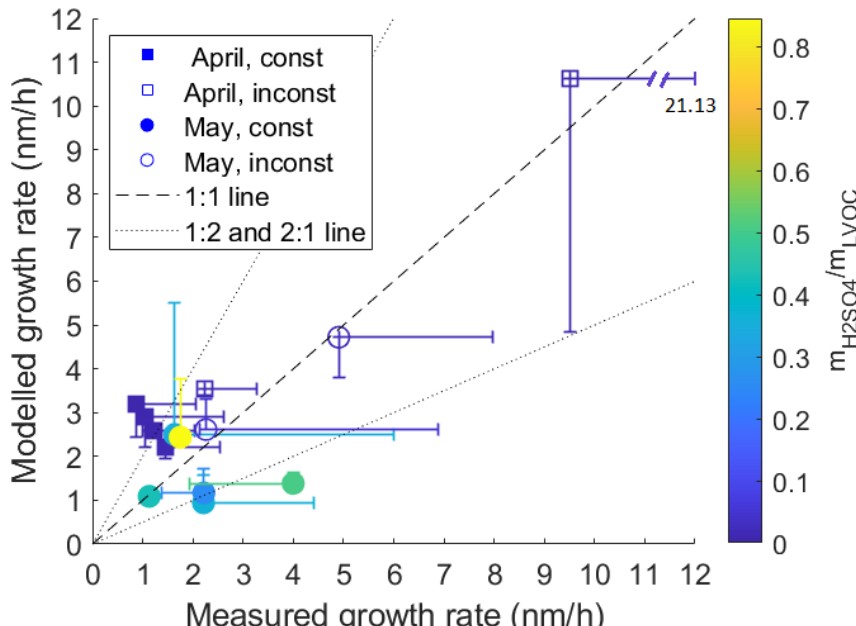

**Figure 1. Modeled vs. measured growth rates of particles. Data from April and May are shown with squares and circles, respectively.**
**The cases where air masses were estimated to be fairly homogeneous throughout the growth event (const) are shown with filled**
**markers and the cases where changes in air mass were assumed to affect the particle size distribution evolution (inconst) are shown**
**with empty markers. The marker color describes the ratio of sulfuric acid and LVOC mass in the simulated particle at the last time**
**point used to define the measured growth rate. The error bars in measured GR represent the different GR obtained using different**
**subsets of nucleation mode peaks during the event. The error bars in the modeled GR represent the GR calculated from the**
**simulation for the same time frame as the corresponding GR from measurements.**

It is also worth noting that the GR in the previous studies were defined using the maximum concentration method (Hirsikko et
al., 2005), i.e., a different method compared to this study. In the maximum concentration method, the GR is determined by a
linear fit to the datapoints of moment of maximum concentration of each size bin, while in the so-called mode fitting GR
method used in this study, the GR is obtained by a linear fit to mode geometric mean diameters determined at each moment.
At the boreal forest site Hyytiälä, the GR values calculated with these two methods had an average relative difference of 16
%, with the maximum concentration method typically producing higher GR values (Yli-Juuti et al., 2011). Therefore, the
difference in GR values obtained in this study and the previous studies for the Chacaltaya site is larger than expected due to
the differences in GR calculation methods. However, the growth rates calculated for high-altitude sites are apparent growth
rates, as the upslope winds and particle transport may affect the values (Sellegri et al, 2019). It is therefore possible that the
two methods are affected differently, and we cannot exclude the possibility that the differences in GR between the studies are
affected by the choice of GR calculation method. In the previous study, Rose et al. (2015) found that the GR at the site are
generally higher during the wet season (November to April) than during the dry season (May to October). The GR determined



in this study, however, showed opposite, with the median GR in May being higher than in April (2.2 nm h$^{-1}$ and 1.1 nm h$^{-1}$,
respectively), albeit the highest GR value, exceeding 9 nm h$^{-1}$, was measured in April. The deviation in our study from the
seasonal difference in GR reported by Rose et al. (2015) can be related to our study taking place at the transition period between
the seasons and including a limited number of NPF events.

For the analyzed events in April (squares in Fig. 1), the model overpredicted the growth, while during May (circles) for some
events the modeled growth was overpredicted and for some events underpredicted. In April, sulfuric acid had very little
contribution to the simulated particle growth (at a maximum 1.2 % of particle mass of a 40 nm particle), whereas in May it
had a large contribution during most events, as shown in Figure 2b. This is in line with the sulfuric acid gas-phase
concentrations being higher during May than during April (average conc. during the analyzed events 6.8 * 10$^6$ # cm$^{-3}$ and 8.3
* 10$^5$ # cm$^{-3}$, respectively) due to occurrence of volcanic emissions in the area (Bianchi et al., 2022; Aliaga et al., 2021). This
increase is illustrated in Fig 2a. The LVOC and ELVOC concentrations on the other hand were lower during May compared
to April (average vapor conc. during analyzed events for LVOCs 2.3 * 10$^7$ # cm$^{-3}$ and 6.0 * 10$^7$ # cm$^{-3}$ and for ELVOCs 7.5 *
10$^5$ # cm$^{-3}$ and 2.1 * 10$^6$ # cm$^{-3}$, respectively). Due to these differences in vapor concentrations, the simulated particle growth
was dominated by LVOCs in April, while in May both LVOCs and sulfuric acid had a major contribution to particle mass.
This is evident from mass fractions presented in Fig. 2b, which show the relative contributions of detected vapors on the
simulated particle mass at 40 nm. It is worth noting that the variability of the growth rate in the simulation and the uncertainty
of the measured GR are large for some events. This is due to the large variability in fitted GR values between the selected
subsets of data points used to define the growth rates, as described in section 2.3, and originates as the consecutive geometric
mean diameters of nucleation mode exhibit deviation from a straight line. For simulated GR, additional uncertainties are caused
by uncertainties in measured organic vapor concentrations, which can be of ± 50 % (Mohr et al., 2019), and saturation
concentrations of organic compounds, which can span over orders of magnitude (O'Meara et al., 2014). The effect of these
possible uncertainties on simulated GR is shown in Figure S4. The modeled growth rate was sensitive for assumed ± 50%
uncertainty in the organic concentrations. This is the case, especially for growth events in April, for which model simulations
shown in Fig. 1 overestimated the GR, while reducing the organic concentrations by 50% decreased the modeled GR to values
lower than the measured GR. For April events the simulations with the assumed upper and lower limit of uncertainty for
organic concentrations led to a large spread in simulated GR. In May, on the other hand, the ± 50% uncertainty in organic
concentration affected the simulated GR less. The difference in the effect of uncertainty of organic concentrations between
April and May is related to the higher organic concentrations in April compared to most events in May. The assumption of ±
50% uncertainty in sulfuric acid concentration had a smaller effect on the simulated growth (Fig. S3).



**Figure 2. a) Mean vapor concentrations of sulfuric acid and organics and b) mass fractions of different compounds in the 40 nm particle in the model simulation for each analyzed day of NPF.**

Figure 3 presents the simulated particle growth compared to the measured evolution of particle size distribution and gas-phase concentrations of organics and sulfuric acid for four example days, April 15 and 22, and May 10 and 27. Similar figures for





all 14 days considered are presented in the supplementary material in Figure S4. The horizontal line visible at 30 nm in Fig. 3 results from combining the data from two instruments, NAIS and MPSS, which were not intercalibrated and compared to each other.

Each panel in Fig. 3 represents one category of days indicated by different markers in Fig. 1 (for rest of the days, see Figure S5). Fig. 3a shows an NPF event during April with relatively homogeneous air mass during the event (filled squares in Fig. 1, 4 days). On these days the gas-phase concentrations of organics stayed relatively constant and sulfuric acid concentrations were low. The measured GR were between 0.9 to 1.4 nm h$^{-1}$ and modeled between 2.1 to 3.2 nm h$^{-1}$. The growth was overpredicted by the model for all these days. The simulated particle growth was dominated by LVOCs, although also ELVOCs

had a clear role (8 to 10 % of the mass in the 40 nm particle), as shown in Fig. 2.

Fig. 3b shows an NPF event during April with changing air mass during the day (squares in Fig. 1, 2 days). The change in air mass can clearly be seen in the particle-phase measurements. Around 10 am to 11 am, within one hour after the NPF event appeared, particle concentrations increased rapidly over a wide size range. Then, in the afternoon, the particle concentrations

decreased rapidly. The changes in organic vapor concentrations were however more subtle and almost no change is seen in the ELVOC and LVOC concentrations. The measured growth rates were 2.2 and 9.5 nm h$^{-1}$, and the modeled GR 3.6 and 10.6 nm h$^{-1}$, respectively, for the two days in this category. For these days, too, the growth was dominated by LVOCs.

Fig. 3c shows an NPF event during May with relatively homogeneous air mass (filled circles in Fig. 1, 6 days). In May, sulfuric

acid concentrations were high and the organic vapor concentrations had some variability during the day. The measured growth rates were between 1.1 to 4.0 nm h$^{-1}$, and the modeled GR between 0.9 to 2.6 nm h$^{-1}$. For four of these six days, the growth was underpredicted (May 13, and 28 to 30) and for two days, overpredicted (May 10 and 22). The main contributors to the growth were LVOCs and sulfuric acid.

Fig. 3d shows an NPF event during May with a change in air mass (circles in Fig. 1, 2 days). On these days, the change in particle size distribution can most clearly be seen in the increase of >100 nm particles. For the organic vapor concentrations, the change is clearly seen as a rapid increase in SVOC and LVOC compounds. The measured GR for these two days were 2.3 and 4.9 nm h$^{-1}$ and modeled GR 2.7 and 4.8 nm h$^{-1}$, respectively. The most important contributor to the growth were LVOC compounds, and unlike the other events in May with constant air mass, sulfuric acid had a negligible effect (see also Fig. 2b).


Considering the relative differences between measured and simulated GR, no differences can be found between different types of days described above. Interestingly, the model runs captured the GR values and changes between days rather well for the days with air mass changes, even though in theory the model is not sufficient to capture the effects of air mass changes on particle composition. For six out of ten of the constant air mass days, the model captured the GR values within a factor of two.



Generally, the results suggest that the detected vapors and their measured concentrations are able to cover a large fraction of the observed particle growth. However, the model did not fully capture the variation between days, and no systematic difference between measured and modeled growth was found. This suggests that some significantly contributing vapors or factors were missing in the model input, i.e., not detected by the instruments. At the same time, due to occasional overestimation of GR, this could also imply the model being too efficient in condensing gases that it has. Alternatively, it may

also be that the detected size distribution was significantly affected by processes not accounted for in the model. For instance, while sudden changes in air mass can be identified from the data, more gradient changes may not be noticed. Such factors may contribute to the differences between the observed and simulated particle GR.

**Table 1. Correlation coefficients (r) and (P) – values of absolute and relative differences between measured and modeled growth**

**rate against different parameters**

| | $GR_{meas} - GR_{model}$ | | $GR_{meas} / GR_{model}$ | |
|---|---|---|---|---|
| | r - value | P - value | r - value | P - value |
| $H_2SO_4$ concentration | 0.21 | 0.47 | 0.10 | 0.73 |
| LVOC concentration | -0.51 | 0.060 | -0.44 | 0.12 |
| ELVOC concentration | -0.75 | 0.0021 | -0.69 | 0.0065 |
| Wind speed | -0.58 | 0.064 | -0.42 | 0.20 |
| Wind direction | 0.66 | 0.028 | 0.76 | 0.0070 |
| Temperature | -0.43 | 0.19 | -0.41 | 0.21 |
| RH | -0.077 | 0.82 | -0.055 | 0.87 |

The difference between simulated and observed growth was further analyzed by calculating correlation coefficients between

the relative and absolute difference of measured and modeled GR and various parameters included in the analysis (i.e. concentration of organics, sulfuric acid, RH, T, wind speed, wind direction, see Table 1 and Figure S6). There was a strong negative correlation for the difference of GR (measured GR – modeled GR) with ELVOC concentration. With low (relative to analyzed days) ELVOC concentrations ($\sim$0.5 $*10^6$ # cm$^{-3}$) the growth was mostly underestimated in the model and with high ELVOC concentrations ($\sim$2$*10^6$ # cm$^{-3}$) the growth was overestimated in the model. With intermediate ELVOC concentrations

the growth was slightly overestimated (Figure S5). Statistically significant negative correlation was also found for the difference of GR values and LVOC concentration, temperature and wind speed, and positive correlation between difference of GR and wind direction. The correlations between the difference in GR values and concentrations of organic vapors may be related to the calibration of the measurement instrument, the uncertainties of saturation concentrations of organic compounds,





or in the fact the instrument did not measure the entire distribution of compounds. However, the model is not sensitive to air

temperature, wind speed and wind direction which are not included in the model at all. This suggests that the underlying reason

for the differences is not necessarily directly dependent on these variables on themselves, but may lie in e.g. changes in air

mass (and hence changes in chemical composition) that our model fails to capture.

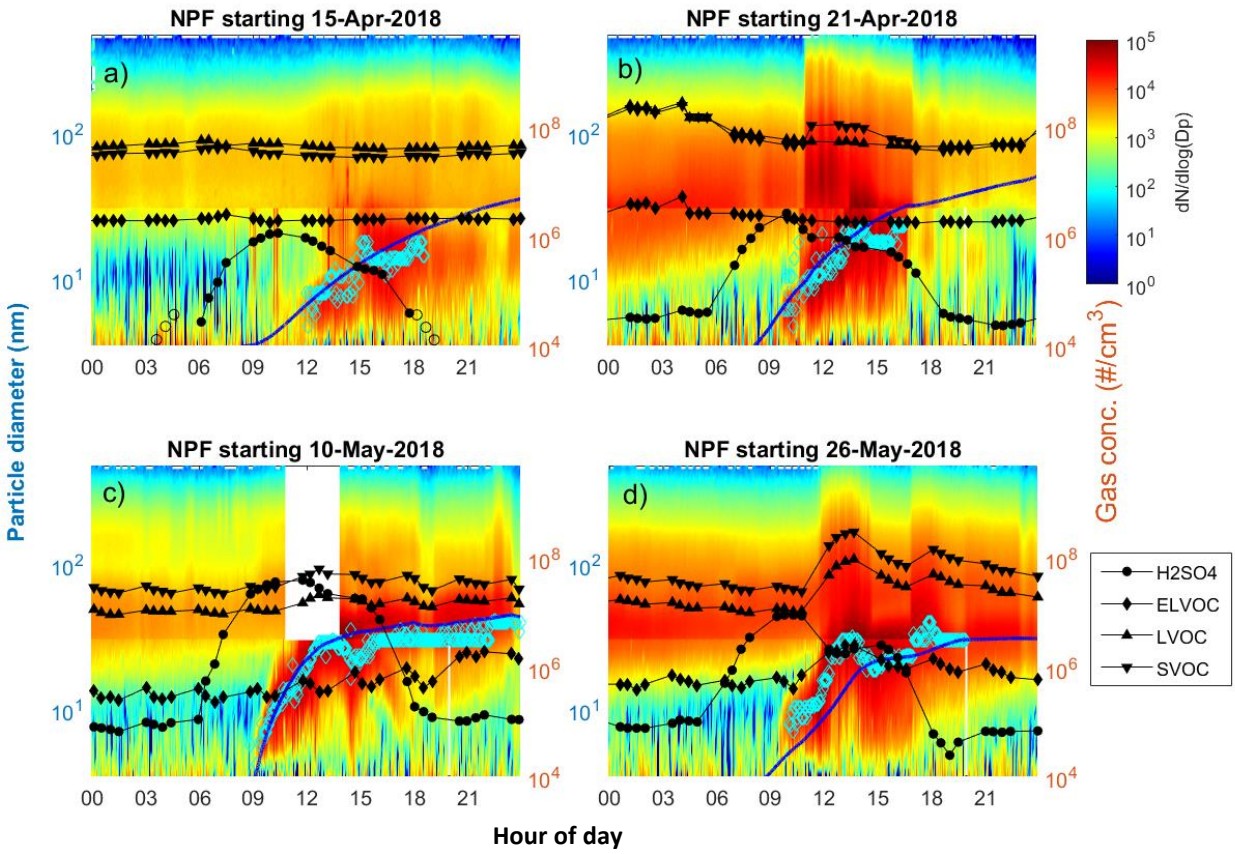


**Figure 3. Modeled particle growth (blue line), fitted geometric mean diameters of nucleation mode (cyan diamonds), measured particle size distribution, and measured gas-phase concentrations of organics and sulfuric acid (black lines with solid markers) as a function of time in a) April with a homogeneous air mass, b) April with an inhomogeneous air mass, c) May with a homogeneous air mass, d) May with an inhomogeneous air mass. The particle diameter for the measured size distribution, modeled particle growth**

**and geometric mean diameters of nucleation mode is on the left y-axis. Gas concentrations are on the y-axis on the right.**

As seen in Fig. 1, cases with an inhomogeneous air mass did not differ significantly from cases with homogeneous air mass in

terms of agreement between measured and simulated particle growth, although overall the GR were larger on days with an

inhomogeneous air mass. In May, the days with inhomogeneous air mass even showed slightly better relative agreement



between measured and modeled GR than most days with constant air mass. However, as discussed above, if the air mass changed during the event, the particles measured in the beginning represent a different particle population compared to the particles measured after the air mass change and, consequently, the GR value calculated from the changes in nucleation mode geometric mean diameter may not represent the real condensation growth rate. The sudden changes in particle size distribution also led to different GR values depending on which nucleation-mode mean diameter data points were included in the

calculation of the GR, which results in large uncertainties as indicated in Fig. 1. Also, the model simulates the growth of one particle from the start of the NPF to the end of the simulation, so even if the model seems to capture a similar behavior as observed in the particle size distribution measurements, they may not represent same processes if the air mass is not homogeneous.

The results show that the size of particles evolving in the background may have had a notably slower GR compared to the overall measured particle population. However, since only the ensemble particle population can be measured, it is hard to make any profound conclusions about how well our model captures the background growth for the cases with inhomogeneous air mass (see Figure S7 for more information).

**4. Conclusions**

We simulated the particle growth during NPF events at the Chacaltaya GAW station based on detected organic vapor and sulfuric acid concentrations for 14 NPF events during April and May 2018. For most events (9 out of 14), the model-simulated GR was within a factor of two of the measurement-based value. According to the model simulations constrained by detected vapor concentrations, the main contributors to the growth were organic vapors, especially the LVOC compounds. However,

during several NPF events, sulfuric acid had a major role, and it contributed in some cases almost half of the simulated particle dry mass at 40 nm. This implies that sulfuric acid from volcanic emissions (Bianchi et al., 2022; Aliaga et al., 2021) also had a notable contribution to the particle growth in the area in May. The effect of sulfuric acid on particle growth at the site is thus expected to be highly dependent on volcanic activity as this effect was not seen in previous studies when volcanoes were less active (Rose et al., 2015).

Our results suggest that the detected vapors and their measured gas-phase concentrations were able to explain a large part of the observed growth. However, based on the differences between measured and simulated growth rates, and particularly due to the low performance of the model in capturing the variation in GR between days, it seems possible that either some significantly contributing vapors were not detected and included in the model, properties of the condensing compounds were not correctly estimated, and/or some other factors, not included in the model, were significantly affecting the growth. The

differences between measured and modeled growth rates might have been affected by the difficulty in determining the growth rate from the particle number size distribution during the campaign. Uncertainty in measured vapor concentrations and estimated saturation vapor concentrations of organic vapors may also contribute to the differences between measured and simulated growth rates. Especially for NPF events in April, the observed nucleation mode growth was within simulations with





assumed ± 50% uncertainty in the organic concentrations. As the growth rate was overestimated in simulations for all of the cases in April, uncertainties in measured vapor concentrations are one potential contributor to differences between model outputs and observations at least for April. In May, the growth rate was both over- and underestimated, and therefore uncertainty in vapor concentrations is less likely an explanation. Due to the differences in the FIGAERO-CIMS instrument setup between April and May, uncertainties in the organic vapor concentrations may have affected results differently between the two months.


These results provide insights into the key factors affecting the particle growth in this high-altitude location, particularly the main vapors condensing onto the nanoparticles, highlighting the importance of low volatile organic compounds and in addition sulfuric acid during volcanic activity periods. The results also demonstrate that the contribution of different vapors on the particle growth may vary between days for this location. However, due to the uncertainties associated with the simulations,

the results should not be interpreted as an exact but indicative representation of the particle growth. Also, the complexity of the surrounding topography, meteorology, and source areas of air masses make it challenging to define whether the measured particle population stays the same during the observed event. Nevertheless, as data on condensing vapors and studies that quantitatively connect these data to nanoparticle growth are still scarce for many atmospheric environments, especially on the Southern Hemisphere, these results contribute to building an overall understanding of NPF and particle growth to CCN sizes.

Due to the indicated dominant effect of organic vapors on the particle growth, the results highlight the importance of improved knowledge of organic vapor properties and further development and application of measurement techniques to identify the wide selection of organic vapors in atmospheric environments.

*Code availability*: The code for MODNAG are available at https://doi.org/10.5281/zenodo.5592258 (Heitto, 2021)

*Data availability:* The datasets are available upon request from the corresponding authors.

*Author contributions:* AH, CW, CM and TY designed the study. AH conducted the simulations. CW, DA, LB, XC, RC, YG, LH, WH, PL, KW, AW, QZ, FB and CM provided observational data. FV, IM, MA provided logistic support for the campaign and meteorological data from Chacaltaya. AH, CW, KL, CM and TY analyzed the results. AH and TY prepared the manuscript

with contribution from all co-authors.

*Competing interests:* The authors declare that they have no conflict of interest.

*Acknowledgments:* This work was supported by the Academy of Finland Center of Excellence programme (grant no. 307331), the Academy of Finland Flagship funding (grants no. 337550 and 337549), H2020 European Research Council (CHAPAs, grant no. 850614) and University of Eastern Finland Doctoral Programme in Environmental

Physics, Health and Biology. We acknowledge the IRD (Institut de Recherche pour le Développement) delegation in Bolivia for assisting with logistics and customs clearance for the instruments and the personnel of IIF-UMSA



for assisting with the station functioning, maintenance and electric troubleshooting. The instrumental deployment at Global GAW station used in this study is supported by an international consortium funded by IRD, Centre National de la Recherche Scientifique (CNRS) and Ministère de la Recherche (under ACTRIS-FR activities), ICOS

Network, Observatoire de Sciences de l'Univers de Grenoble (under Labex OSUG@2020), Leibniz Institute for Tropospheric Research, Consejo Superior de Investigaciones Científicas and University of Stockholm.

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
