# Peer review of "Analysis of atmospheric particle growth based on vapor concentrations measured at the high-altitude GAW station Chacaltaya in the Bolivian Andes"

_EGUsphere, 2023_

## Author Response (AR1)

Answers to reviewer comments regarding the manuscript "Analysis of atmospheric particle growth based on vapor concentrations measured at the high-altitude GAW station Chacaltaya in the Bolivian Andes"

Comments by the reviewers are written below on **bold**, our reply in normal text and modifications for the manuscript in *italic*.

Additional change outside of reviewer comments:

In the original manuscript there was a typo. In the results section and in Figure 2 it was stated that the results presented are for a particle with diameter of 40 nm. However, in reality the results presented are for 30 nm particle. This change does not affect our conclusions anyway. The particle diameter has now been corrected in lines 33, 341, 350 and 368.

RC 1

**The manuscript is well ready for discussion. The reviewer has a few minor comments for the authors considering. The measurements of CCN are critical to confirm which chemicals drove the growth of newly formed particles. However, the data were not included here while the station had taken the measurements. Anything happened? How about the growth of pre-existing particles? Does the result support the analysis presented here?**

We thank reviewer for their comments. Regarding the CCN measurements, no direct CCN measurements have been made at Chacaltaya and hence they are not included in our analysis. Rose et al. (2017) did report CCN calculations, but they were based on SMPS data, not CCN counter data.

Related to the pre-existing particles, they are not directly considered in our model study. However, since our model is constrained by the measured vapor concentrations, which are affected also by condensation on the pre-existing particles, the effect of growth of pre-existing particles to the vapor concentrations is indirectly taken into account in the model.

RC 2

**This work reports an analysis of aerosol particles growth from both measurements and modelling approaches. It aims to grow the knowledge on new particle formation events in high altitudes in the southern hemisphere. It uses a complex and extended dataset during the SALTENA campaign, and the results could fit within the scope of ACP, being of interest for the international research community. The manuscript is well organized and written, however, at the current status, I would recommend this manuscript to be published as a measurement report. If not, the manuscript needs to provide more significant advance for aerosol science before it is published as research article in ACP, and I would suggest some aspects to be considered in order to improve the manuscript and/or strengthen its impact.**

We thank the reviewer for the thorough review of our manuscript and for the comments that helped us in improving the manuscript. We have considered all the comments and modified the manuscript as needed. Below we reply to the comments point-by-point. Regarding the suggestion of publishing our manuscript as a measurement report instead of a research article, we find that our study doesn't merely report new interesting measurement results, but substantially advances our general understanding of atmospheric compounds contributing to the particle growth. As of now, most of the CIMS measurements reported are from near the sea level, and it is highly valuable that in this study the perspective has been expanded to higher altitudes. In addition, since only few direct particle-phase composition measurements of atmospheric particles are available and they have large uncertainties, there is a demand for methods to estimate particle phase composition, In this study we present rather simple model approach to do these estimations based on gas phase measurements, that usually are better available. Using this method in our study, our substantial findings are the variability of the contributing vapors to the particle growth between days and months and especially the significant role of volcanic activity to the particle growth in volcanically active areas. Hence we argue, that this study should indeed be published as a research article. We have also now emphasized these points in the Abstract of the manuscript where we have rephrased the text in lines 26-32:

*"Despite the challenging topography and ambient conditions around the station, simple particle growth model used in the study was able to show that the detected vapors were sufficient to explain the observed particle growth, although some discrepancies were found between modelled and measured particle growth rates. This study, one of the first of such studies conducted on high-altitude, gives an insight on the key factors affecting the particle growth on the site and helps to improve the understanding of important factors on high-altitude sites and the Atmosphere in general. Low volatile organic compounds originating from multiple surrounding sources such as Amazonia and La Paz metropolitan area, were found to be the main contributor to the particle growth, covering on average 65% of simulated particle mass in particle with diameter of 30 nm. In addition, sulfuric acid had a major contribution to the particle growth, covering at maximum 37% of simulated particle mass in 30 nm particle during periods when volcanic activity was detected on the area, compared to around 1% contribution on days without volcanic activity. This suggests that volcanic emissions can greatly enhance the particle growth. "*

**Major comments**

1. **The main objective of combining models and measurements is to validate and/or improve models. However, in this case, the model does not really provide insights on the uncertainties because, as stated by the authors, at mountain sites the comparison of modeled and real GR provides the comparison of different processes (homogeneous vs non-homogeneous conditions). How the results of this work can be used in future works? Which are the uncertainties of the model if you are directly comparing processes that are different (homogeneous vs non-homogeneous)?**

   The main objective of this study was to study how different factors (most importantly organic vapors and sulfuric acid) contribute to the initial secondary aerosol growth on this high-altitude site and if the measured vapors can explain the detected particle growth. On one hand the study tests the ability of the model, which is based on gas-particle transportation without including particle phase processes, in capturing the observed growth. However, even more importantly, the study explores the connection between observed vapor

concentrations and particle growth, and thus tests and validates the measurements of condensable vapors.

It is true that our model assumes homogenous conditions and as stated in the manuscript caution must be used when the model results are compared to the measurements made in inhomogeneous conditions. However, in this manuscript we present also multiple events with relatively homogeneous conditions and for these days our model approach gives valuable information on the factors affecting initial particle growth.

2. **One of the major limitations that the reader can appreciate to apply this model in real atmosphere is that precursor vapors are only consumed to form/grow new particles, but what about the larger particles or condensation sink?**

In our study the gas phase concentrations are not affected at all by the model. Instead, the model uses measured gas phase concentrations which already are affected by condensation sink. Therefore, while larger particles are not directly included in the model, their effect on vapor concentrations is indirectly included.   To emphasize this, we have now added to the line 168 this sentence:

*"In this study measured vapor concentration of organic vapors and sulfuric acid were used as an input. It is worth noting that by doing this the reduction of ambient vapors by condensation sink is also indirectly taken into account in the model."*

3. **I recommend the authors to extend the study, e.g. by analyzing the contribution of different vapours to different size ranges (GR 3-7, 7-20,…), comparing different models (e.g. MALTE-BOX or UHMA models),… Despite this work uses state-of-the-art instrumentation, there is no significant advances. If there is not additional results, I would suggest this manuscript to be published as a measurement report.**

To our understanding, the models mentioned by the reviewer are box models. Therefore, the use of them, or other aerosol box models, would not provide significant benefit to the analysis presented here. While a box model that includes full particle size distribution might allow better analyzing size dependent particle composition compared to MODNAG model, they would still be inherently constrained by the same assumptions as MODNAG (homogenity of air mass) and we argue that comparing them would not give us any additional information. We have chosen to not analyze the size dependent contributions of vapors here, and instead concentrate on the overall picture of the role of vapors from different volatilities. Detailed analysis of the size dependence from the model simulations would also increase the sensitivity of the results to assumed vapor properties, and, with the fluctuations in size distribution at this site creating challenges to the determination of particle growth rate for specific size ranges, the validation data from observations would be limited.

Multiple CIMS measurements have been reported in the literature, but still we lack measurements from many relevant environments. In addition, information on what organic compounds are important for particle growth in different environments still have big uncertainty. Hence, we argue, that our study in this high-altitude Southern Hemisphere location does advance our general knowledge of role of organic compounds in early growth of atmospheric particles.  We were able to show that within uncertainties the measured vapor concentrations on the spot are sufficient to explain the observed particle growth. We

also confirmed the major role of SVOCs and sulfuric acid on particle growth, along with smaller but still notable contribution of ELVOCs. We believe that our study lays a good ground work for further research with e.g. other models.

**Minor and technical comments**

**L25 – space before dot**

We have removed the space.

**L29 – Dot after "40 nm"?**

We have removed the dot.

**L42-43 – what about evaporation?**

We have added now sentence *"Assuming the gas phase concentrations are high enough to allow condensational growth of the particles,"* to the text to emphasize that we are focusing on situations where the net effect of condensation and evaporation to the particle growth is positive.

**L44-45 – this sentence is not completely correct, rephrase. If particles continue growing, will act as CCN, but if it collides with pre-existing particles doesn't mean it will not act as CCN…**

We have now rephrased the text to:

*"The growth of particles serves as a source of CCN-sized particles while their collisions with pre-existing larger particles reduces the production of potential CCN from the secondary aerosol."*

**L46 – this reference is probably not the best, some more recent could better support the role of H2SO4 on nucleation (e.g. Sipilä et al., 2010; Ehn et al., 2014).**

We thank the reviewer for the suggested citations. We have added Sipilä et al 2010 citation.

**L70-72 – There is previous studies that discussed this phenomenon and its effect on NPF events. For example, Garcia et al. (2014), Foucart et al. (2018) and Casquero-Vera et al. (2020) already showed the appearance of particles in a large range of diameters at the same time that NPF starts and noticed the difficulties to identify and characterize NPF events at mountain sites. Specially Foucart et al. (2018) mentioned this process and the term "apparent" growth rates at mountain sites because of the advection, non-homogeneous air masses and the overestimation of GR. I think these works need to be cited here or along the manuscript because, in addition to Sellegri et al. (2019), these studies have pointed some of the discussed phenomena.**

Thank you for pointing out these relevant studies. All of them have now been added to the citation list in line 74

**L76 – "is influenced"**

As the text is edited (see response to the comment below), this typo is removed.

**L76-77 – how can the station be at free troposphere and boundary layer at same time? Actually, next sentence doesn't give a percentage for both occurring at same time. Maybe change by "could be simultaneously affected by long- and short-range aerosol and precursor vapors transport"?**

Thank you for the suggestion to make the text clearer! We have now edited the text accordingly and now it reads:

*"CHC can be simultaneously affected by long- and short-range aerosol and precursor vapors transport (Aliaga et al., 2021)."*

**L80-86 – why giving these values in the introduction? I would recommend removing this lines from the introduction.**

We think these values given here are relevant to give understanding of conditions in the area and put our study in context.

**L92 – "the role that different precursors/organic compounds…"?**

We have now rephrased the line as:

*"the role that different vapors play in the growth"*

**L139-142 – what about the absolute or relative differences in ELVOCs region? The scale doesn't allow the comparison in this region (Fig. S1).**

Thank you for pointing this out. We have now changed the y-axis in Figure S1 to logarithmic scale to ease the comparison. However, regarding our study the ELVOC concentrations from Nitrate CIMS are generally so low that including them would not make large impact on the results.

**L261 – point and parenthesis**

Typo is corrected.

**L261-263 – the authors state again that there are multiple air masses affecting the station, if that the case, how can you compare real NPF events with modelling (e.g. modelling consider homogeneous conditions as it is mentioned at the beginning of this paragraph)?**

As discussed in our answer to the first major comment of the reviewer, it is true that one has to be cautious when comparing measured and modelled data in this kind of complicated conditions. This is why we have brought it up and discussed it in our manuscript. However, we also still argue that on many of the events presented in our study the conditions are homogeneous enough to allow meaningful and insightful comparisons between measured and modelled data.

**L267-271 – this doesn't completely agree with the results presented by Rose et al. 2017, who showed that there is a clear pattern of the boundary layer (BL) at CHC. Is then the BL and FT conditions the same air mass?**

The difference in results of these two studies comes from different methods used in them to identify air masses, wind direction in Rose et al. and FLEXPART modelling in Aliaga et al. However, in the end both articles agree that air mass with high vapor concentrations and particle concentrations originates from the PBL.

**L314-315 – what this sentence means?**

We thank reviewer for noticing this unclear sentence. We have now clarified the sentence by changing it to:

*"Therefore, the difference in obtained GR values between this study and the previous studies for the Chacaltaya site is larger than would be expected from solely the differences in GR calculation methods"*

**Fig. 1 - why error bars are only in one direction?**

For most of the events, determining which mode peaks should be included to GR calculations was not straightforward (for example determining when the event ends). Hence, for each of such an event, multiple GR calculations were conducted with slightly different selection of mode peaks. The calculation which best described the event was chosen as the main GR calculation and the rest are presented as an error bar in Fig. 1 In modelled GR the error bars represent GRs calculated from the modelled particle diameter evolution over similar time frame as in respective GR calculation from measurements. In all events the GR calculated from the main set of mode peak diameters and the respective GR from model happen to give lowest or highest GR value from all subsets and hence the error bar goes only in one direction. The explanation of what the error bars represent in the Fig. 1 is given in Fig. 1 captions in line 309-311.

**L341 – how sensitive is the model to the initial particle number concentration? According to the example provided, it is fixed to 2000 #/cm3 with diameters of 2nm?**

As MODNAG simulates only single particle, the initial particle concentration is not an input variable for the model and is not sensitive to it at all.
We assume the reviewer refers to the MODNAG.m file in the GitHub repository. We thank the reviewer for being so throughout when going through the codes in the repository and noticing this total particle number concentration value. However, this value is not used in MODNAG, total number concentration as an input is a remnant from older code that was used when making the code for MODNAG analysis. The value is now removed from the code to prevent future confusion. In addition, it is also important to notice that the results presented in this manuscript are not created with MODNAG.m but with MODNAG_Bolivia.m code. MODNAG.m was used in our previous paper (Heitto et al., 2022). Unfortunately, we had the wrong DOI number in the manuscript that was related to the other MODNAG version. We have now updated the DOI number in the manuscript.

**L356 – please include in the Fig. S4 the corresponding GR as text in each subplot (the values used for Fig. 1) and the geometric diameters (as Fig. 3).**

We have modified the Fig S4 as suggested.

**L374&380 – Change "Fig. 3x" by "Figure 3x"**

We have changed these in lines 373, 380 and 386.

**Figure 3 – Local time? Please also indicate in the text when you refer to the events time.**

The reviewer is correct, the presented times are in local time. We have added the mentioning of this in the caption of Figure 3 and also included the mention in line 231 by adding:

*"(onwards presented as local time, UTC-4)"*

**L440-444 – here the authors states about the growth of larger particles but actually the model does not include the size distribution of to simulate the growth of those pre-existing particles, so here probably refers to already "formed" from the model? Please clarify this paragraph.**

We thank the reviewer for this comment. In this paragraph by "background" we meant freshly formed particles that are measured in the morning and that assumingly continue growing during the day over the area, but which we cannot measure, since the air mass at the station changes. We did not refer to the pre-existing larger particles. We understand that our choice of term may have caused confusion and we have now rephrased the paragraph as follows:

*"The results show that the size of particles evolving in the air mass detected in the morning and again possibly in the afternoon may have had a notably slower GR compared to the overall measured particle population. However, since only the ensemble particle population can be measured, it is hard to make any profound conclusions about how well our model captures the growth in the air mass measured in the morning for the cases with inhomogeneous air mass (see Figure S7 for more information). "*

**L485 – The code does not work, please verify the files provided.**

> **Initial radius of the particle is 1e-09 m.**
>
> **Undefined function or variable 'flx_eaim_MABNAGO_hd_pd_AH'.**
>
> **Error in odearguments (line 90)**
>
> **f0 = feval(ode,t0,y0,args{:});   % ODE15I sets args{1} to yp0.**
>
> **Error in ode15s (line 150)**
>
> **odearguments(FcnHandlesUsed, solver_name, ode, tspan, y0, options, varargin);**
>
> **Error in MODNAG (line 235)**
>
> **[tout1, output1] = ode15s(@flx_eaim_MABNAGO_hd_pd_AH,time,input(3:end),options,whats,diss_frac,RxnP, DP);**

Thank you for pointing this out. We apologize for the inconvenience. There was a mistake in the code where an old name of differential equation solver function was used in the code instead of the new updated one. The codes in the repository are now updated and should work correctly.

**References**

**Casquero-Vera, J. A., Lyamani, H., Dada, L., Hakala, S., Paasonen, P., Román, R., Fraile, R., Petäjä, T., Olmo-Reyes, F. J., and Alados-Arboledas, L.: New particle formation at urban and high-altitude remote sites in the south-eastern Iberian Peninsula, Atmos. Chem. Phys., 20, 14253–14271, https://doi.org/10.5194/acp-20-14253-2020, 2020.**

**Ehn, M., Thornton, J., Kleist, E., Sipila, M., Junninen, H., Pullinen, I., Springer, M., Rubach, F., Tillmann, R., Lee, B., LopezHilfiker, F., Andres, S., Acir, I., Rissanen, M., Jokinen, T., Schobesberger, S., Kangasluoma, J., Kontkanen, J., Nieminen, T., Kurten, T., Nielsen, L., Jorgensen, S., Kjaergaard, H., Canagaratna, M., Dal Maso, M., Berndt, T., Petaja, T., Wahner, A., Kerminen, V., Kulmala, M., Worsnop, D., Wildt, J., and Mentel, T.: A large source of low-volatility secondary organic aerosol, Nature, 506, 476–479, doi:10.1038/nature13032, 2014.**

**Foucart, B., Sellegri, K., Tulet, P., Rose, C., Metzger, J.-M., and Picard, D.: High occurrence of new particle formation events at the Maïdo high-altitude observatory (2150 m), Réunion (Indian Ocean), Atmos. Chem. Phys., 18, 9243–9261, https://doi.org/10.5194/acp-18-9243-2018, 2018.**

**García, M. I., Rodríguez, S., González, Y., and García, R. D.: Climatology of new particle formation at Izaña mountain GAW observatory in the subtropical North Atlantic, Atmos. Chem. Phys., 14, 3865–3881, https://doi.org/10.5194/acp-14-3865-2014, 2014.**

**Sipilä, M., Berndt, T., Petaja, T., Brus, D., Vanhanen, J., Stratmann, F., Patokoski, J., Mauldin, R., Hyvarinen, A., Lihavainen, H., and Kulmala, M.: The role of sulfuric acid in atmospheric nucleation, Science, 327, 1243–1246, doi:10.1126/science.1180315, 2010.**

References for author's respond

Heitto, A., Lehtinen, K., Petäjä, T., Lopez-Hilfiker, F., Thornton, J. A., Kulmala, M., and Yli-Juuti, T.: Effects of oligomerization and decomposition on the nanoparticle growth: a model study, Atmos. Chem. Phys., 22, 155–171, https://doi.org/10.5194/acp-22-155-2022, 2022.

---

## Author Response (AR2)

Authors' reply to the reviewer's comments. Comments by the reviewer are written below in **bold**, our reply in normal text and modifications for the manuscript in *italic*.

One general modification to the manuscript is made, we have changed the text of competing interests (L512) to "At least one of the (co-)authors is a member of the editorial board of Atmospheric Chemistry and Physics" according to the remarks by the editor.

**Despite the authors have answered the reviewer concerns, the reviewer still have a major concern about one of the general comments.**

First, this is an editorial decision, but I would like to point that, despite we agree there is a demand of studies on this topic, this work applies a model (already published before) to data (already published before; e.g., Zha et al., 2024) in order to model growth rates (considering vapour properties already used before) and compare them with measured ones. The authors stated that "We also confirmed the major role of SVOCs and sulfuric acid on particle growth, along with smaller but still notable contribution of ELVOCs", however this is not a new finding because this comes from applying the model (this model only depends on vapours concentrations and not on pre-existing particle concentrations) to already published data. In this sense, despite this manuscript provides new useful information, from my point of view, results and conclusions of this manuscript are of more limited scope than in research articles.

The Southern hemisphere high ALTitude Experiment on particle Nucleation And growth (SALTENA) campaign has been notably fruitful, resulting in several publications (Zha et al., 2023b; Scholz et al., 2022; Zha et al., 2023a; Bianchi et al., 2022). These studies, derived from different observational periods and using different instruments, address diverse research topics. Notably, the gas-phase organic compounds detected by the FIGAERO-CIMS have not been covered in previously published papers from this campaign.

Zha et al. (2023b) only published particle-phase FIGAERO-CIMS data for only one day, while our study used gas phase FIGAERO-CIMS data. While that paper focuses on the long-distance transport of oxidized compounds (with 4-5 carbon atoms) in the tropical free troposphere air from Amazonia, our study primarily investigates new particle formation and growth in the boundary layer. While Zha et al. (2023b) focuses on the measurement in January 2018 during the austral summer (wet season), we reported the NPF events in April (the wet-to-dry transition period) and May (austral winter, dry season) 2018. In Zha et al. (2023b), the measurement was performed with a nitrate CI-APi-TOF that predominantly measures ELVOCs. For comparison, it presented in the SI one event during the night of 22 April 2018. In this figure, only the mass fraction of the C4-C5 organic compounds in the gas-phase (again with the nitrate CI-APi-TOF) and particle-phase (with the FIGAERO-CIMS) was presented. In our manuscript we present and use the volatility distribution of the measured gas-phase organic vapors, which is an approach not published before from the SALTENA campaign. We would also argue, that by looking gas-phase measurements alone it would not be possible to determine the important compounds affecting particle growth. More detailed analysis is needed, which is provided in this manuscript by MODNAG model and volatility information of organic precursor vapors.

In our manuscript we compared the modelled particle growth and the observations to test whether the measured organics and sulfuric acid are enough to explain the observed particle growth, which, clearly, is a new finding. The reviewer states that our conclusion is not new because it comes from applying the model (which only depends on vapour concentrations and not on pre-existing particle concentrations) to already published data. Firstly, as explained below, it doesn't need to include all processes affecting the gas-phase

concentration as input data. Secondly, we don't see any concerns regarding the loss of novelty when we applied a well-established and published model to our unpublished data. Hence, we still argue that our manuscript contains enough novelty to be published as a research article rather than as a measurement report and that it contributes to advance our understanding on components taking part in particle growth on high altitudes.

Lastly, we would like to apologize for a typo in our last reply. In sentence "We also confirmed the major role of SVOCs and sulfuric acid on particle growth, along with smaller but still notable contribution of ELVOCs", we meant LVOCs to have a major role in the particle growth, not SVOCs, whose contribution we showed to be almost negligible. This typo was only in our reply for the reviewer comment and in the manuscript (on line 381) this had been written correctly.

However, the reviewer main concern is about the second and third major comments. The authors stated that "model uses measured gas phase concentrations which already are affected by condensation sink". To my knowledge, gas phase concentrations are affected by 1) condensation sink and 2) formation and growth rates. Actually, the precursor vapours are consumed by these three processes 1) condensation in pre-existing particles, 2) formation of new particles and 3) the growth of newly formed particles. Thus, the reviewer here don't really understand the argument that the authors provide about this comment. Why are concentrations only affected by condensation? Furthermore, I agree that "the size dependence from the model simulations would also increase the sensitivity of the results to assumed vapor properties" but if the condensation into pre-existing particles is high, then the comparison with the model is not accurate at all.

The reviewer is correct that all these processes, i.e., formation and growth of the new particles and condensation to larger pre-existing particles, affect the precursor vapors. However, our reasoning still applies, i.e., we only need the measured gas phase concentration for each simulation steps, instead of calculating the changes of gas phase concentration due to multiple source and sink processes for the next simulation step. By using the measured gas phase concentrations through the simulation, we are simulating how a particle would grow in that ambient surrounding gas phase (manuscript line 174-176). Important point to note here is that we do not use the measured gas phase concentration in the model only as an initial value at the time when the particle starts to grow. Instead, in our model for each time step in the simulation we use real-time and time-dependent gas-phase concentration measurements as an input. The measured gas phase data with 30 min resolution was interpolated to the time resolution required by the model for this (manuscript line 220-222). These measured gas phase concentrations have been affected by the processes mentioned by the reviewer and by any other processes that may affect the concentrations. As gas-phase concentrations are taken at each time step from measured data, the changes in gas-phase concentrations are not calculated in the model and are thus not dependent on calculated condensation rate. Hence, although condensation sink to particles of any size or particle formation are not directly considered in our model, they are indirectly counted for by using measured gas-phase concentrations of condensing vapors. This is one of the advantages of our rather simple model, that can capture the growth of newly formed particles despite other factors affecting precursor vapors.

**Finally, about the minor comment of Fig. 1, I still have same question, why error bars are only in one direction? I understand the process that the authors have followed, however, if you have applied different tests or retrievals, why the "good" GR is not the mean/median and is instead one of the "extreme" values?**

It is important to notice that the error bars in our figure do not describe an error in any statistical sense, but different GRs obtained from different subsets, (this is explained also in the manuscript on line 318-319). Usually, when this method to calculate the GR is used, only one set of mode peak diameters is selected and used for determining the GR for the event (e.g., Dal Maso et al., 2005; Nieminen et al., 2014). This method

includes a step where the mode peak diameter points to which the line is fitted are selected. For instance, the mode fitting may result sometimes mode peak diameters that do not describe the growing modes properly and thus need to be excluded from the fitting of the line to obtain GR. At this point the researcher makes the decision of which datapoints to include in the line fitting for the GR based on visual inspection. The markers in our Fig. 1 describe the GR values which are obtained from the set of mode peak diameter data points that were evaluated by eye to describe the event best, i.e., these correspond to how GR values with this method are typically reported. However, as we have stated in our manuscript, in our study most of the events were not "smooth" enough to straightforwardly define one set of mode peak diameters that would perfectly describe the event) that seemed to possibly also describe the growing mode and fitted a line to also these, obtaining one to three additional GR values that may describe the growth. To be exact, together there were 14 event days for wich GR was calculated, of which for one day we only selected one set of mode peaks, for six days two subsets and for the rest seven days tree to four subsets.

The main reasons for selecting the addition subsets of mode peak diameters for GR calculation were 1) the difficulty in determining which time period (how long) one should consider for the GR line fitting and 2) the challenge in selecting the correct mode peaks data points to describe the growth when there was fluctuation or e.g. two consequent events during a day. In our previous reply to the reviewers we wrote by mistake that the "best" subset gave always lowest or highest GR. We apologize for this. For two out of the seven events with more than two subsets the "best" subset gives the smallest GR and also for the remaining five cases the GR for the "best" subset is very close to the smallest GR value (difference < 0.3 nm/h), due to which the error bar is mostly not visible under the marker in Fig. 1. When looked closely this can be seen in one of the events (lower of the not-filled circle markers in the Fig. 1). Out of the six events with two subsets, there are two events where the "best" subset gives largest GR (filled blue and green circles at the bottom of the Fig. 1), and four events where the "best" subset gives lowest GR. Altogether this means that there are 11 events where the "best" subset gives lowest or close-to-lowest GR value and only two events where it gave the largest GR value out of the different GR fits for the event. This is reasonable, since the "best" subset was always also the subset that was obtained over longest time period and the rest were taken from some shorter parts of the event. When selecting a shorter period of the event, one is less likely to include periods when growth was so slow that the diameter seems to be changing little or not at all, and thus such shorter period tends to give higher GRs compared to a longer period. For ten of the events one of the chosen subsets was the first monotonic growth period of mode peaks. From this subset the obtained GR was often quite different compared to other subsets, which were relatively similar to each other.

As described in the manuscript on lines 254-258, the "best" subset describes our best estimation of the particle growth and with the error bars we have aimed to highlight the uncertainty of this estimation, due to the "unsmoothness" of the events. We argue that using mean or median value in these cases would give too much weight on the extreme value (now described by the large error bars in the figure) and not describe the best estimate of the GR properly.

References

Bianchi, F., et al., Bulletin of the American Meteorological Society, 103, E212-E229, 2022.
Scholz, W., et al., EGUsphere, 2022, 1-42, 2022.
Zha, Q., et al., Atmos. Chem. Phys., 23, 4559-4576, 2023a.
Zha, Q., et al., National Science Review, 2023b.
Dal Maso et al., Boreal Env. Res. 10: 323–336, 2005.
Nieminen et al., Boreal Env. Res., 19, Suppl. B, 191–214, 201